# Rapid SERS Detection of Botulinum Neurotoxin Type A

**DOI:** 10.3390/nano13182531

**Published:** 2023-09-10

**Authors:** Alexei Subekin, Rugiya Alieva, Vladimir Kukushkin, Ilya Oleynikov, Elena Zavyalova

**Affiliations:** 1Moscow Institute of Physics and Technology, 141701 Dolgoprudny, Russia; alexey.subekin@gmail.com (A.S.); ruqiwa_eva@mail.ru (R.A.); 2Chemistry Department, Lomonosov Moscow State University, 119991 Moscow, Russia; 3Osipyan Institute of Solid State Physics of the Russian Academy of Science, 142432 Chernogolovka, Russia; kukushvi@mail.ru; 4A.N. Belozersky Institute of Physico-Chemical Biology, Lomonosov Moscow State University, 119991 Moscow, Russia; oleynikov.biophys@gmail.com

**Keywords:** aptamer, aptasensor, biosensor, botulinum neurotoxin, SERS

## Abstract

Surface-enhanced Raman scattering (SERS) is a powerful technique for decoding of 2-5-component mixes of analytes. Low concentrations of analytes and complex biological media are usually non-decodable with SERS. Recognition molecules, such as antibodies and aptamers, provide an opportunity for a specific binding of ultra-low contents of analyte dissolved in complex biological media. Different approaches have been proposed to provide changes in SERS intensity of an external label upon binding of ultra-low contents of the analytes. In this paper, we propose a SERS-based sensor for the rapid and sensitive detection of botulinum toxin type A. The silver nanoisland SERS substrate was functionalized using an aptamer conjugated with a Raman label. The binding of the target affects the orientation of the label, providing changes in an analytical signal. This trick allowed detecting botulinum toxin type A in a one-stage manner without additional staining with a monotonous dose dependence and a limit of detection of 2.4 ng/mL. The proposed sensor architecture is consistent with the multiarray detection systems for multiplex analyses.

## 1. Introduction

Botulinum neurotoxins are produced by *Clostridium botulinum*, an anaerobic Gram-positive bacterium, the causative agent of botulism. Four of the seven serotypes of botulinum neurotoxins (BoNT) can cause botulism in humans (A, B, E, and F); among them, serotypes A and B are the most common. 

Along with anthrax, plague, tularemia, and viral hemorrhagic fevers, BoNT is classified as Category A by the US Center for Disease Control and Prevention [1,2,3]. BoNT has a highly lethal effect. Its LD_50_ (lethal dose in 50% of cases) upon inhalation is 10–13 ng/kg inhaled, 1.3–2.1 ng/kg intravenously, and 1 mg/kg orally [4]. These data were obtained for the two most common types of toxin A and B (BoNT A and BoNT B). This article focuses only on BoNT A. According to [5], 1 g of dispersed BoNT in aerosol form can kill up to 1 million people in a densely populated area. Since it can be obtained relatively easily with artisanal means, BoNT is attractive as a weapon of mass destruction in bioterrorist acts. Its application in present-day cosmetic medicine as a component of established medical drugs allows potential terrorists to easily disguise and transport large quantities of the purified toxin.

Botulinum neurotoxins are 150 kDa proteins and zinc-dependent proteases (Figure 1). Receptor-binding domains vary among the serotypes targeting specific SNARE proteins on the cell’s surface. Toxins are activated with translocation into the cell using the translocating domain; afterwards, the catalytic domain is separated from other domains and blocks the release of neurotransmitters. This mechanism of the action of BoNT is used in medicine and cosmetology to treat pain, excessive sweating, drooling, and for wrinkle reduction. The most popular medicals, Botox, Xeomin, and Dysport, are from serotype A [6].

The only FDA-approved method for BoNT detection, the so-called “gold standard”, is based on the determination of drug toxicity in mice [8,9]. Despite the high sensitivity of this method (about 10–20 pg/mL), it is very time-consuming (about 7 days) and does not meet present-day requirements to ensure national security and human health. Therefore, different variants of BoNT rapid detection are being developed at present.

Surface-enhanced Raman spectroscopy (SERS) is one of the most promising methods that meet modern requirements for rapidity, sensitivity, and specificity of indications of various toxins. This method has already found application in different fields of analytics, including the detection of hazardous chemical and biological agents [10]. Currently, various options for BoNT detection using SERS have already been proposed.

Lim et al. proposed a kind of heterogeneous sandwich solid-phase immunoassay of BoNT A and BoNT B [11]. For specific capture of the antigen from the analyte, authors proposed to use slides with immobilized gold nanoparticles and specific antibodies on the toxin. An “external Raman label” (ERL) was applied to detect BoNTs. The ERL is a gold nanoparticle with BoNTs-specific antibodies and 5,5′-dithiobis(succinimidyl-2-nitrobenzoate) reporter molecules immobilized on its surface. The limit of detection (LoD) of BoNT A in the proposed system was 0.7–1.2 ng/mL with the analysis time not exceeding 1 h.

Kim et al. suggested another variant of the “sandwich” using magnetic particles with capture antibodies [12]. The authors utilized nanotags as a detecting agent, which, similar to the previous work, are gold nanoparticles with toxin-specific antibodies and malachite green reporter molecules immobilized on them. The LoD of the proposed system was about 5.7 ng/mL BoNT with an analysis time not exceeding 2 h.

The immunoassay options mentioned above demonstrate the high potential of SERS in the rapid detection of BoNT A. However, the use of the detectors mentioned above has several limitations. The techniques for obtaining ERLs and nanotags are complex and difficult to reproduce. This can significantly affect the reproducibility of toxin determination between different batches of detection particles, as the authors do not provide statistics for multiple bunches of ERLs or nanotags. The use of antibodies in such detectors has several limitations due to the labor-intensive nature of their production and poor stability at room temperature. Alternative approaches can be used to substitute antibodies in the recognition process, e.g., SERS-based sensors with molecularly imprinted polymers [13] or aptamers [14]. In this work, the specific detection of BoNT A is implemented using aptamers. 

Nucleic acid aptamers are recognizing elements based on the structured oligonucleotides. Contrary to antibodies, aptamers are developed in vitro using a combinatorial approach called SELEX (Systematic Evolution of Ligands by EXponential Enrichment) which is remarkably convenient when working with toxic substances. Moreover, aptamers are synthesized chemically with the possibility of extensive modification with labels, anchors, and other functional groups for sensor creation [14]. Several aptamers have been selected for BoNTs. Lou et al. [15] described several DNA aptamers to the recombinant light chain of BoNT serotype A (i.e., catalytic domain of the protein) with dissociation constants (K_D_) in the range of 34–86 nM. Tok and Fischer [16] developed several DNA aptamers for BoNT serotype A (aldehyde-inactivated full-length protein) with high affinity (K_D_ values were in the range of 3 to 51 nM). Oh et al. [17] developed DNA aptamers to the recombinant light chain of BoNT serotype C with superior affinity (K_D_ values were in the range of 0.7 to 43 nM). Furthermore, RNA aptamers were selected by Chang et al. [18] for the recombinant light chain of BoNT serotype A with K_D_ in the range of 90 to 190 nM. Dissociation constants of aptamer-BoNT complexes in the nanomolar span allow the creation of biosensors with as low LoD as 0.04 ng/mL with a time of analysis of 24 h [19].

Here, we describe a simple design of the aptasensor, where the SERS-surface-anchored aptamer serves as a recognition molecule and Raman label simultaneously. This idea allows detection of BoNT A in the binary complex with the aptamer without sandwich assembly and additional staining procedures. The approach was previously used in our work to detect whole viruses (influenza A, respiratory syncytial virus, adenovirus, and severe acute respiratory syndrome-related coronavirus 2) [20]. Large targets reoriented Raman labels on the surface, providing an analytical signal. Here, we demonstrate the applicability of this approach to medium targets, such as individual proteins, aiming to widen possible applications for the one-stage process.

## 2. Materials and Methods

### 2.1. Reagents

Inorganic salts, sucrose, Tween 80, human serum albumin, and gamma globulin (IgGHum) (AppliChem GmbH, Darmstadt, Germany and Sigma–Aldrich, St. Louis, MO, USA) were used in this work. A commercial set of bovine blood serum samples was used (Biolot, Saint Petersburg, Russia).

All experiments were performed in ultrapure water prepared with Milli-Q equipment (Merck KGaA, Darmstadt, Germany). PBS was prepared from tablets from EcoService (Saint Petersburg, Russia). HEPES buffer, pH 7.5, from AppliChem (Darmstadt, Germany), 1-ethyl-3-(3-dimethylaminopropyl)-carbodiimide hydrochloride (EDC-HCl) from Roth (Karlsruhe, Germany), and sodium salt of N-hydroxysulfosuccinimide (s-NHS) from Chem-Impex Int’l (Wood Dale, IL, USA) were used.

Xeomin 50 Units per vial (Merz therapeutics, Frankfurt am Main, Germany) contains BoNT serotype A in human serum albumin with sucrose. The powder was dissolved in 125 µL Ringer’s solution (Grotex, Saint Petersburg, Russia) providing a solution with 400 Units/mL BoNT, 8 mg/mL human serum albumin, and 38 mg/mL sucrose. The solution was exploited immediately after the preparation. According to Frevert [21], 400 Units/mL of Xeomin corresponds to 1.76 ng/mL or 11.7 pM of BoNT serotype A.

### 2.2. Oligonucleotides

The oligonucleotides were synthesized using the solid-phase phosphoramidite method by a commercial company (Syntol, Moscow, Russia). The sequences and modification sites are presented in Table 1. Aptamer conformation was assembled in 1 μM aptamer solution in PBS buffer (pH 7.3) with heating at 95 °C for 5 min and cooling at room temperature.

### 2.3. Affinity Assay

The affinity of the aptamers to BoNT was estimated using biolayer interferometry (Blitz, ForteBio, Menlo Park, CA, USA). The experiments were conducted at 20 °C. Samples were placed in black 0.5 mL tubes (Sigma–Aldrich, St. Louis, MO, USA) with 220 μL of volume. Biosensors intended for the amine coupling reaction (Octet AR2G biosensors, ForteBio, Menlo Park, CA, USA) were hydrated for 10 min in water. The sensors were activated for 5 min in the solution of 200 mM EDC-HCl and 100 mM s-NHS. Then, BoNT was loaded from a 10 µL drop of 1.76 ng/mL solution. The sensor was washed and blocked with 100 mM HEPES-HCl buffer with pH 7.5 for 3 min. After the signal stabilization in the PBS, the association step was conducted with aptamer solution in the concentration of 1000, 500, or 250 nM in PBS buffer. Then, the dissociation step was conducted in a PBS buffer. Both association and dissociation stages were monitored for 200 s. DNA aptamer to streptavidin, Apt_2 (see Table 1), was used as a reference sample for subtraction of non-specific interactions.

### 2.4. Circular Dichroism and UV Spectroscopy

A 1 μM aptamer solution in PBS buffer with pH 7.3 was placed in quartz cuvettes with a 1 cm path. Circular dichroism (CD) and UV spectra were acquired using the CD spectrometer Chirascan (Applied Photophysics, Leatherhead, Great Britain) and equipped with a thermoelectric temperature regulator. The spectra were acquired in the wavelength range of 240–360 nm. The spectrum of the buffer was subtracted at baseline. The samples were heated with a mean ramp of 1.0 °C/min. The melting experiments were conducted in the range of 10–90 °C. The melting temperatures were derived from the temperature dependencies of UV absorption at 260 nm and molar circular dichroism at 247 and 270 nm.

### 2.5. SERS Substrates 

Two nanostructured silver zones were deposed on a silicon plate covered with a 300 nm thick layer of SiO_2_ using the thin film deposition system NANO38 (Kurt J. Lesker Company, Jefferson Hills, PA, USA). Silver deposition with a nominal thickness of 6 nm was performed at a rate of 0.4 A/s. After sputtering, the substrate with two active SERS zones was annealed at 240 °C for 6 min on an HP-20D-Set heating plate (Daihan Scientific, Gang-Won-Do, Republic of Korea). As a result, these zones were covered with silver nanoislands with an average planar size of 24 nm and a dispersion of 14 nm. One of the zones was used as the experimental zone, and another zone was the control.

### 2.6. Aptamer Immobilization

We applied 10 µL of 140, 70, 35, 17.5, and 8.8 nM Apt_2.22_mod aptamer solution in PBS (pH 7.3) to the experimental and control zones. Droplets were incubated in sealed Petri dishes for 1 h; then, they washed sequentially with PBS containing 0.01% Tween 80. 

### 2.7. Reaction with the Antigen

We placed 10 μL of 2.4 ng/mL BoNT A into PBS containing 52 mg/mL sucrose and 11 mg/mL human serum albumin in the experimental zone. PBS with sucrose and albumin of the same concentrations was used as a control. The reaction was carried out in sealed Petri dishes for 1 h; then, it was washed consecutively with PBS containing 0.01% Tween 80, PBS, and deionized water. The substrates were dried at room temperature and then SERS was measured.

### 2.8. SERS Measurements

The SERS spectra were obtained using an optical scanning microscope Olympus BX51 (Olympus Corporation, Tokyo, Japan) based on the RamanLife RL532 spectrometer (TeraSense Group, Inc., San Jose, CA, USA) with a laser wavelength of 532 nm and output power of 5 mW. The spectrometer had a spectral resolution of 4–6 cm^−1^ and a spectral range of 160–4000 cm^−1^. The diameter of the laser spot was 10 µm. A distribution map (2 mm × 2 mm) of the SERS signal from Cy3 over the entire area of the sample was recorded on a 10x objective lens with a step of 200 microns in XY-scanning mode. The maps of the distribution of the intensity of the Raman peak 1645 cm^−1^ were recorded. Statistical data were obtained on ten SERS substrates for each point. Fifty measurements were obtained for each substrate. The average values and measurement errors for all measurements were calculated.

### 2.9. Study of Surface Topology

Scanning electron microscopy (SEM) images were obtained using the Scanning Electron Microscope Supra 50VP (Zeiss, Oberkochen, Germany). The electron optical GEMINI column provides excellent beam brightness with an ultrahigh resolution of 1 nm at the accelerating voltage of 20 kV. Substrate surface and profiles were scanned at the accelerating voltage of 10 kV, with an aperture size of 30 µm, at the work distance of 8 mm, and at the chamber pressure of 9 × 10^−4^ Pa.

## 3. Results

### 3.1. Selection of the Aptamer for the Biosensor

Two DNA aptamers with the highest affinity to BoNT serotype A were chosen from the literature, namely aptamers 2.22 selected by Tok and Fischer [16], and aptamer C selected by Lou et al. [15] (Table 1). RNA aptamers were not included in the series due to their high cost compared to DNA aptamers. Also, aptamers without reported K_D_ values were not studied. Only one of two aptamers, Apt_2.22, formed a highly affine complex with Xeomin (Figure 2). The stability of the complex is considered to be the most significant property of the aptamer in biosensing applications. Apt_2.22 dissociated slowly from the BoNT-modified sensor that perfectly fits the requirements to recognize molecules in biosensors. The K_D_ was estimated to be 7 ± 4 nM, which is close to the previously reported value of 3 nM [16]. Apt_C dissociated from the BoNT-modified sensor in a minute due to having K_D_ > 100 nM; therefore, it was excluded from further research.

The thermal stability of the recognizing elements is one more essential property for biosensing applications. The aptamer has to be stable in the operating range of the sensor, which commonly lies in the span of 20–37 °C. The secondary structure of aptamer Apt_2.22 was predicted using an RNAfold [23] (see Figure 3C). The aptamer consists of three duplex regions connected with loops. The absence of non-canonical nucleic acid structures can be noticed in the CD spectrum (Figure 3A); the spectrum has a negative maximum at 247 nm, which is characteristic of DNA duplexes [24]. Melting experiments were conducted simultaneously for CD and UV spectroscopies, giving the same melting temperatures, 38.3 ± 1.4 °C and 38.4 ± 0.4 °C, correspondingly (Figure 3). Thus, this aptamer has moderate thermal stability, and its folded fraction exists below 25 °C. Further experiments were performed at a temperature of 20 °C.

### 3.2. Determination of the Optimal Aptamer Concentration

The idea of the sensor for BoNT A detection presented in this article is based on its specific interaction with the aptamer, causing a decrease in the signal from the label. At sufficiently low concentrations of antigen in the analyte, the difference between the signal values of the experiment and control may be relatively small. Therefore, the optimal aptamer concentration was determined using two criteria:Narrow distribution of SERS intensity, providing good reproducibility of experimental results;The average SERS intensity value that provides the lowest possible BoNT A detection limit.

The results are presented in Table 2 and Figure 4.

According to the data presented in the table, the aptamer concentration of 35 nM is optimal in terms of the coefficient of variation (only 0.06), but, at the same time, the average signal value is rather high—19,300 a.u.—which may affect the sensitivity of the sensor. Therefore, three Apt_2.22_mod solutions with concentrations of 35, 17.5, and 8.8 nM were taken for further experiments. Figure 5 demonstrates the distributions of SERS values on the substrate. The Raman peak of 1645 cm^−1^ is inherent in the TAMRA label, which is conjugated with the aptamer. This peak was chosen for processing because of the maximal plasmon absorption of the SERS substrate in this spectral region [25].

### 3.3. Detection of BoNT A

We performed experiments with a BoNT A. To detect the toxin, we evaluated the ratio of SERS signal in the experimental and control zones. In the case of its specific binding caused by the aptamer, the signal intensity from the experimental zone decreased compared to that in the control zone. Experiments with statistically significant differences were considered positive. The results are presented in Table 3.

Detection of BoNT A at a concentration of 2.4 ng/mL is statistically significant at a concentration of Apt_2.22_mod 17.5 nM (with a 95% probability at *p* < 0.001, see Table 3). The distributions of SERS intensities in the experimental and control zones are shown in Figure 6B. We also conducted an experiment to determine the BoNT A detection limit (Figure 6A). SERS_cont_/SERS_exp_ ratio depends on the concentration of BoNT A. The concentration of the toxin containing 1.2 ng/mL is determined in no more than 50% of cases. The detection limit of BoNT A for the sensor is 2.4 ng/mL since the values of SERS intensity for this concentration do not overlap with the value of SERS intensity in the control zone.

### 3.4. Choosing the Optimal Analysis Time

It should be noted, that for low concentrations of BoNT A, such as 2.4 ng/mL, if the exposure time of the antigen on the surface of the aptamer sensor is too short, the probability of false negatives increases. We performed a separate experiment in which we varied this parameter. The results are shown in Figure 7.

According to the data in Figure 6, the normalized SERS signal is increased with the increase in incubation. The detection time of BoNT A is 60 min and provides statistically significant differences between experiment and control samples (the difference exceeds 3 standard deviations). Therefore, the incubation time of 60 min was chosen for all subsequent experiments.

### 3.5. Defining the Specificity of BoNT A Binding on SERS Substrate

Similar experiments were conducted with an off-target substance, human immunoglobulin G (IgGHum). This protein was chosen due to its mass being almost the same as that of BoNT A—150 kDa. Furthermore, considering the prospect of using our proposed sensor in clinical diagnostics, this protein will be one of the most “interfering” components in blood plasma analysis. The detection results are shown in Table 4.

The data in Table 4 reveal that the ratio of the SERS signal in the experimental and control zones is statistically equal. This indicates the absence of non-specific binding of IgGHum and of the aptamer utilized. The distributions of the intensity of the signals of the experiment and control are provided in Figure 8.

We also performed experiments with BoNT-non-specific aptamer. The DNA aptamer for the human respiratory syncytial virus strain A2 was originally developed by Percze et al. [26] and further modified with the thiol group and TAMRA dye [20]. The results are displayed in Table 5.

Similar to the previous experiment, we showed that in the case of the replacement of Apt_2.22_mod with an aptamer non-specific to BoNT A, a positive result in toxin detection was not observed. The distribution of SERS intensities is presented in Figure 9. The usage of off-target analyte or aptamer did not provide positive results; therefore, these experiments proved high affinity and specificity of Apt_2.22_mod to BoNT A.

### 3.6. The Influence of the Matrix on BoNT A Detection

Bovine blood serum was used to estimate the influence of a mix of off-target substances in BoNT A detection. BoNT A at a concentration of 2.4 ng/mL was added to the serum samples, and toxin-free serum was used as a control. The results of the experiment are presented in Table 6 and Figure 10.

BoNT A was determined as statistically reliably in serum (95% probability at *p* < 0.001) at a concentration of 2.4 ng/mL. Thus, the complex matrix has no practical effect on the detection of the toxin. The sensor detects BoNT A in both simple and complex mixtures with different contents of proteins, low-molecular substances, carbohydrates, etc.

The shelf life of our sensors is limited only by SERS substrates. It was found previously that the signal intensity of the test substance on this type of substrates is halved after 6 months of storage. The substrates used are disposable, but inexpensive in manufacturing (the estimated cost of one sensor is USD 0.5). These properties are compatible with the practical implementation of these sensors.

## 4. Discussion

Recently, technologies for the specific detection of various substances using aptamers in combination with surface-enhanced Raman spectroscopy have been actively developed. Noticeable progress in the detection of various biological targets such as protein toxins [27], bacteria [28], and viruses [29] was achieved. At the same time, one of the modern requirements for such sensors is the development of multiplexed platforms that enable the simultaneous detection of several targets. Therefore, the optimization of the structure of SERS platforms and the concentration of detecting aptamers to ensure high analysis sensitivity and resolution concerning each detectable substance remain relevant issues.

In this paper, we reported a SERS substrate developed in an inexpensive, reproducible, and simple technology that operates efficiently under 532 nm of laser excitation. An aptamer-based sensor designed on this structure can determine botulinum toxin type A with a limit of detection of 2.4 ng/mL in 1 h, which is consistent with existing antibody-based analogs for BoNT A with LoD of 0.7–1.3 ng/mL within 1 h of analysis [11] and 5.7 ng/mL within 2 h of analysis [12]. The aptamer-based electrochemical biosensors provided 50-fold lower LoD of 0.04 ng/mL with a substantially higher time of analysis (1 day instead of 1 h) [19]. The statistical data showed good reproducibility of experimental results and high specificity of BoNT A detection for a new sensor. The particular components of human blood plasma, namely serum albumin and immunoglobulin G, were tested to estimate non-specific binding that could provide false-positive results. Both proteins did not interfere with the detection of BoNT A; therefore, the primary experiments revealed no cross-specificity. Moreover, a complex matrix (bovine blood serum) did not interfere with toxin detection. Further testing of real biological fluids is necessary for the practical implementation of these sensors. 

The primary benefit of the described aptasensor is its simplicity. The nanoisland SERS substrates are rather cheap (cost of production does not exceeding USD 2), with a scalable production technique. The DNA aptamers are stable during storage; therefore, the sensors could be assembled and packed long before the experiments. The whole assay is maximally simple, namely the binary complex is assembled, washed, and SERS-signal is acquired. We suppose the LoD could be reduced further by replacing the Apt_2.22 with K_D_ = 7 nM and with an aptamer with K_D_ in the picomolar range. Previous SERS-based sensors for BoNT A detection used multistep procedures, namely assembly of sandwich-like complexes with antibodies, external Raman labels, nanotags, etc. [11,12]. One-step techniques are much easier in implementation, but signal enhancement in a binary complex is the main obstacle. A novel approach was proposed for virus detection in multiplex SERS-based aptasensors [20]. The Raman label is conjugated with the recognizing element immobilized on the SERS substrate. The binding of the target affects the orientation of the label on the metal surface providing changes in the analytical signal. Conformational changes in aptamers are often used in sensing; for example, FRET-based (Förster resonance energy transfer-based) aptasensors provide an analytical signal due to changes in distance between two aptamer regions during analyte binding [30,31]. The proposed aptasensor has a similar recognition mechanism that was used successfully for protein detection using SERS-based techniques.

The suggested aptamer-based sensor provides direct one-step detection of BoNT A that is consistent with the multiarray detection systems for multiplex analysis in a sample. Further optimization of Raman systems will allow obtaining of the multiplex systems for sensitive detection of several targets. Such sensors may find applications in medical diagnostics of various diseases as well as in the field of national security.

## Figures and Tables

**Figure 1 nanomaterials-13-02531-f001:**
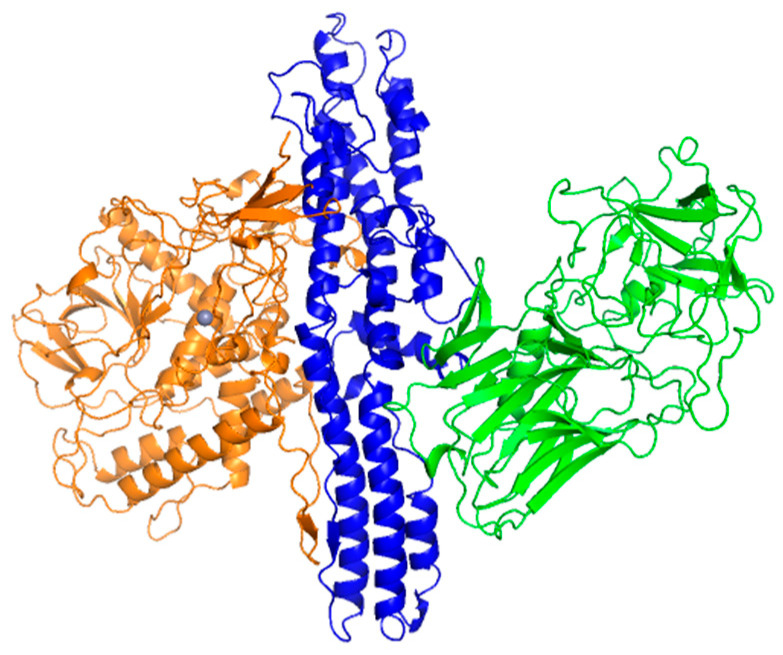
Structure of botulinum neurotoxin serotype A. Receptor-binding domain is shown in green color; the translocating domain is shown in blue color; the catalytic domain is shown in orange color. The structure was built using PyMol software (2.4.1 version) from the crystal structure with pdb id 3BTA [7].

**Figure 2 nanomaterials-13-02531-f002:**
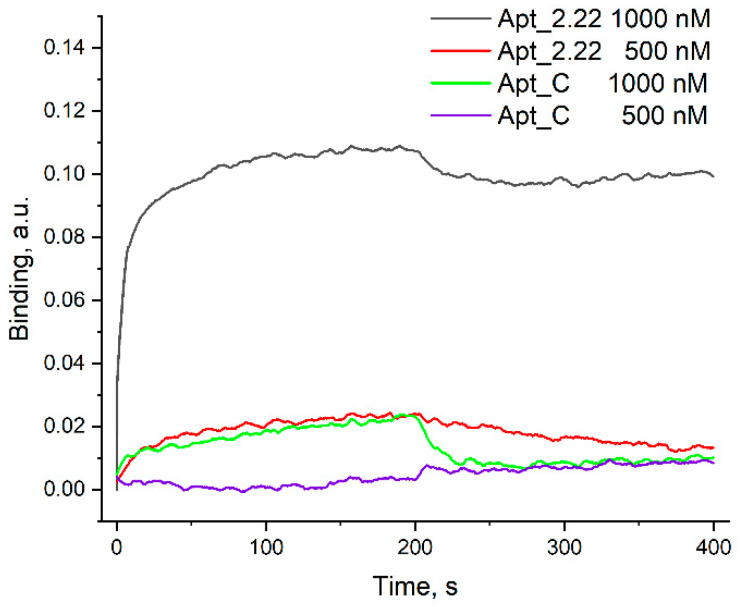
Affinity of DNA aptamers to BoNT serotype A (Xeomin) estimated with biolayer interferometry. The association stage corresponds to 0–200 s of the experiment; the dissociation stage corresponds to 0–200 s of the experiment.

**Figure 3 nanomaterials-13-02531-f003:**
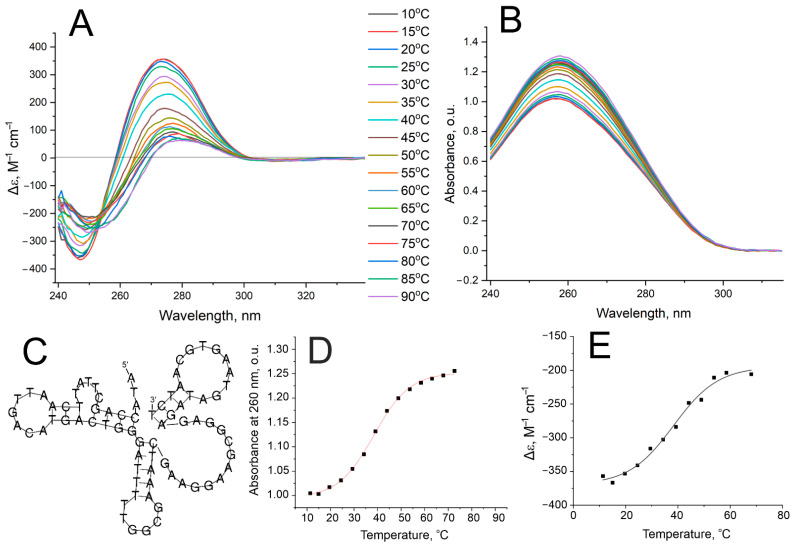
Thermal stability of aptamer Apt_2.22 estimated with CD and UV spectroscopies. CD (**A**) and UV spectra (**B**) are provided at different temperatures. The predicted structure of the aptamer is shown in part (**C**). Melting curves are shown for the UV melting experiment at a wavelength of 260 nm (**D**) and CD melting experiment at a wavelength of 247 nm (**E**).

**Figure 4 nanomaterials-13-02531-f004:**
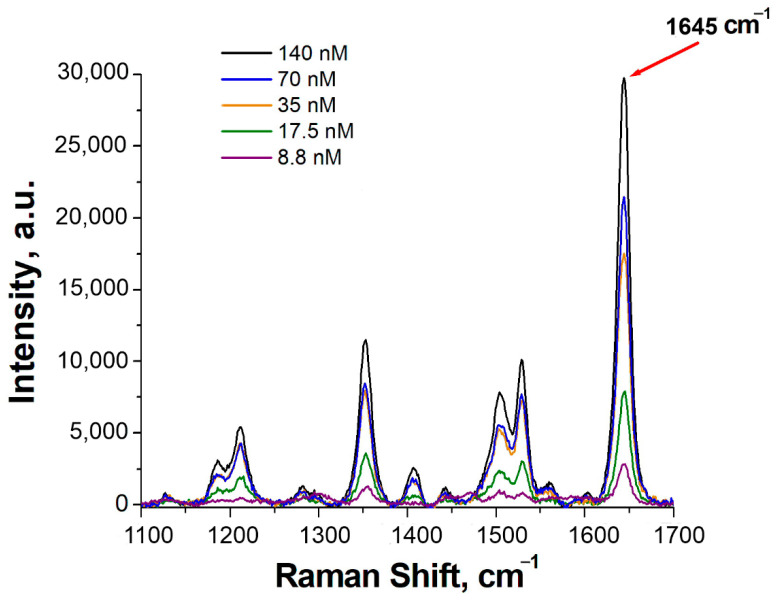
SERS spectra of the substrates obtained with different concentrations of Apt 2.22_mod solution.

**Figure 5 nanomaterials-13-02531-f005:**
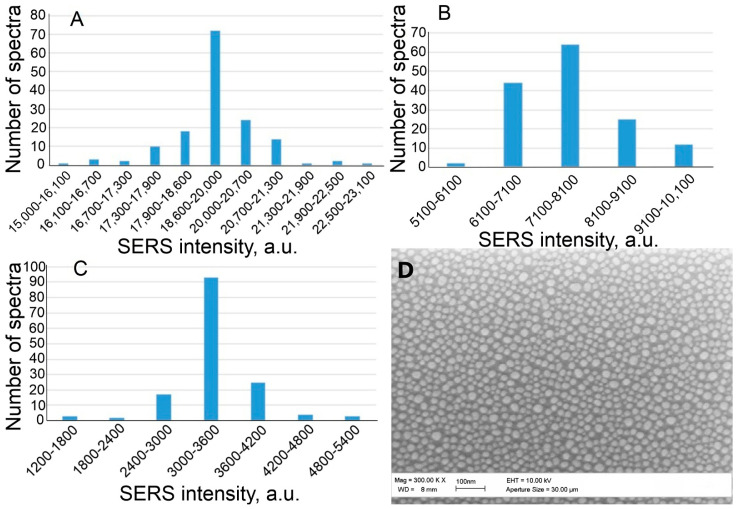
Distribution of SERS intensity at Apt_2.22_mod aptamer concentration of 35 nM (**A**), 17.5 nM (**B**), and 8.8 nM (**C**). Part (**D**)—SEM image of SERS substrate.

**Figure 6 nanomaterials-13-02531-f006:**
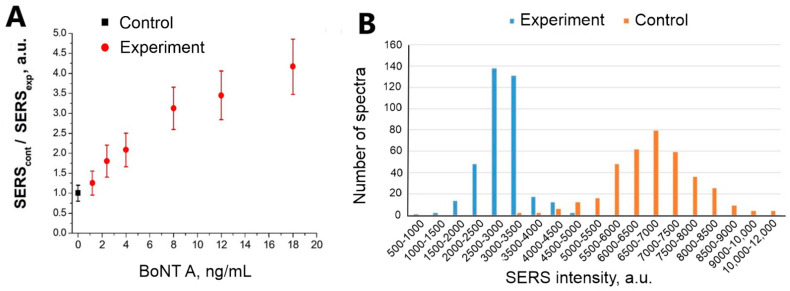
(**A**) The dependence of SERS signal on BoNT A concentration. SERS substrates were functionalized using Apt_2.22_mod at the concentration of 17.5 nM. (**B**) Distribution of SERS signal intensities on experimental and control zones during BoNT A (2.4 ng/mL) detection.

**Figure 7 nanomaterials-13-02531-f007:**
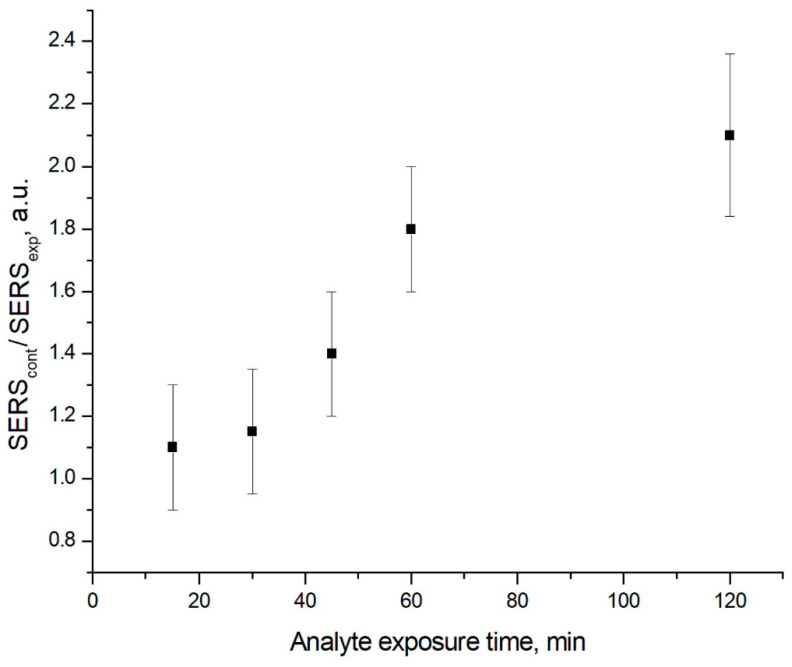
Dependence of SERS signal on the analyte exposure time (BoNT A).

**Figure 8 nanomaterials-13-02531-f008:**
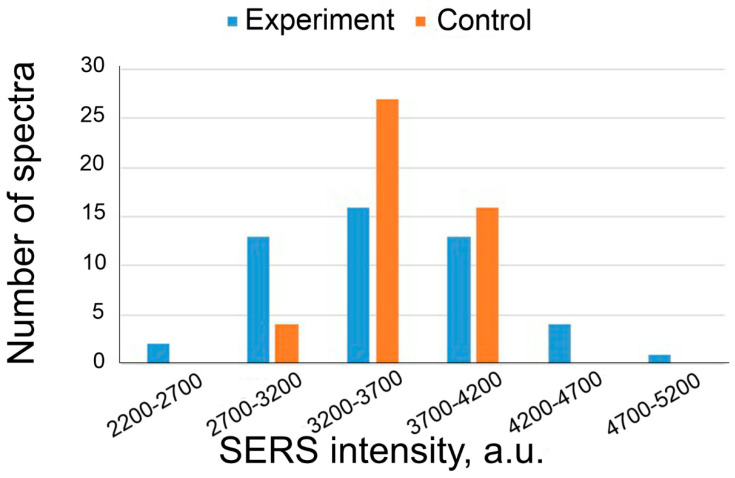
SERS intensity distributions of experiment and control while detecting IgGHum (2.4 ng/mL) using Apt_2.22_mod aptamer at a concentration of 17.5 nM.

**Figure 9 nanomaterials-13-02531-f009:**
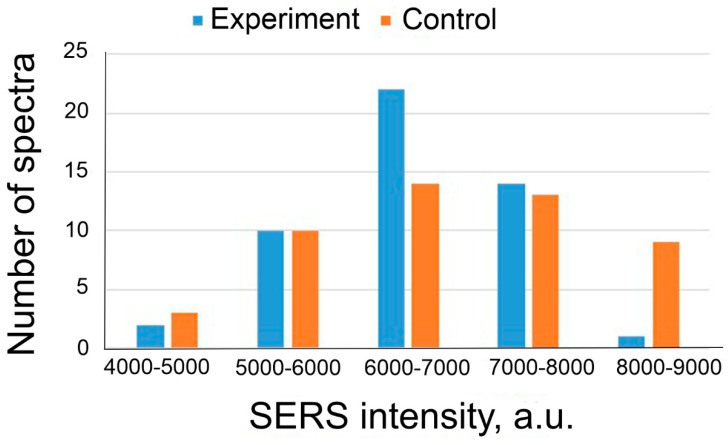
SERS intensity distributions in the samples with 2.4 ng/mL of BoNT A (experiment) and the control samples. SERS substrates were functionalized with aptamer of RSV at a concentration of 17.5 nM.

**Figure 10 nanomaterials-13-02531-f010:**
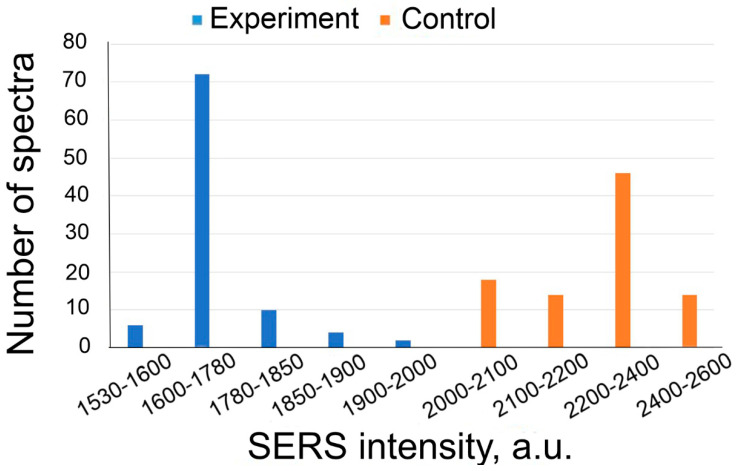
SERS intensity distributions in the samples of blood serum with 2.4 ng/mL BoNT A (experiment) and toxin-free blood serum (control).

**Table 1 nanomaterials-13-02531-t001:** Sequences of DNA aptamers studied in this work. Dissociation constants (K_D_) are provided from the original works; n.d.—not determined.

Code	Sequence	Length, Nucleotides	K_D_, nM	Reference
Apt_2.22	5′-ATACCAGCTTATTCAATTGACATGACTGGGATTTTTGGCG AAATCGAAGGAAGCGGAGAGATAGTAAGTGCAATCT-3′	76	3	Tok and Fischer (2008) [16]
Apt_C	5′-AGCAGCACAGAGGTCAGATGCTTGAGTGTCATGGACGTT CCGGTCTTGGGCGGGATATTTGTTTGTTTTCTGCCTATGTTC CTATGCGTGCTACCGTGAA-3′	100	34	Lou et al. (2009) [15]
Apt_2	5′-AGCAGCACAGAGGTCAGATGAGGTTTAGTGAATATCTTC GATGATCCGAGGCAGGCTAGATTCCGAAACATCGCTGAGC GCCTATGCGTGCTACCGTGAA-3′	100	No	Oh et al. (2009) [22]
Apt_2.22_mod	(SH-(CH_2_)_6_)-5′-ATACCAGCTTATTCAATTGACATGACTGGGATTT TTGGCGAAATCGAAGGAAGCGGAGAGATAGTAAGTGCAA TCT-3′-(TAMRA)	76	n.d.	This work
SH-H8-TAMRA	(SH-(CH_2_)_6_)-5′-AGTGCGGTGAGCCGTCGGACATACAAATAC-3′-(TAMRA)	30	No	Kukushkin et al. (2023) [20]

**Table 2 nanomaterials-13-02531-t002:** Results of the experiment on the determination of the optimal concentration of Apt_2.22_mod solution.

Aptamer Concentration, nM	Average Value of Signal Intensity (μ), a. u.	Variation Coefficient (V)
140	30,000 ± 12,000	0.39
70	23,000 ± 7000	0.32
35	19,300 ± 1200	0.06
17.5	7600 ± 900	0.12
8.8	3300 ± 500	0.16

**Table 3 nanomaterials-13-02531-t003:** Detection of BoNT A.

Concentration of Apt_2.22_mod Aptamer, nM	Intensity Value (Experiment), a.u.	Intensity Value (Control), a.u.	Probability of Determining BoNT A (Concentration 2.4 ng/mL), % (*p* < 0.001)
35	16,000 ± 2000	16,000 ± 2000	0
17.5	2900 ± 600	6500 ± 1200	95
8.8	2700 ± 1200	2000 ± 600	0

**Table 4 nanomaterials-13-02531-t004:** Detection of IgG_Hum_.

Concentration of Apt_2.22_mod Aptamer, nM	Intensity Value (Experiment), a.u.	Intensity Value (Control), a.u.	Probability of Non-Specific Binding IgG_Hum_ (Concentration 2.4 ng/mL), % (*p* < 0.001)
17.5	3500 ± 600	3500 ± 200	No more than 2.5

**Table 5 nanomaterials-13-02531-t005:** Detection of BoNT A using BoNT-non-specific aptamer.

Concentration of SH-H8-TAMRA Aptamer, nM	Intensity Value (Experiment), a.u.	Intensity Value (Control), a.u.	Probability of Non-Specific Binding BoNT A (Concentration 2.4 ng/mL), %
17.5	6400 ± 400	6100 ± 600	No more than 2.5

**Table 6 nanomaterials-13-02531-t006:** Determination of BoNT A in blood serum.

Concentration of Apt_2.22_mod Aptamer, nM	Intensity Value (Experiment), a.u.	Intensity Value (Control), a.u.	Probability of Determining BoNT A (Concentration 2.4 ng/mL), % (*p* < 0.001)
17.5	1700 ± 100	2200 ± 200	95

## Data Availability

Data is contained within the article.

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
