# Peer review of "Rapid SERS Detection of Botulinum Neurotoxin Type A"

_nanomaterials, 2023, doi:10.3390/nano13182531_

Round 1
Reviewer 1 Report
This is an exciting work; however, before proceeding to the next step, the authors should address the following comments.
1. Provide a more in-depth comparison and discussion of related previous works.
2. Authors should also provide more meaningful discussions regarding the repeatability and reproducibility of the conducted tests/analysis.
Author Response
Q.1. Provide a more in-depth comparison and discussion of related previous works.
A.1. We are grateful for the evaluation of our work. We expanded the Discussion part to address the first comment. The following sentences with additional refs were added:
‘Previous SERS-based sensors for BoNT detection used multistep procedures, namely, sandwich-like complexes with multiple components like antibodies, external Raman labels, nanotags [11,12]. One-step techniques are much easier in implementation, but signal enhancement in binary complex is the main obstacle. Here we used a novel approach that was proposed for virus detection in multiplex SERS-based aptasensor [20]. The Raman label is embedded in the recognizing element that anchored to the SERS substrate. Binding of the target affects the orientation of the label on the metal surface providing changes in analytical signal. Conformational changes in aptamers are often used in sensors, e.g. FRET-based (Förster resonance energy transfer-based) aptasensors provide an analytical signal due to changes in distance between two aptamer regions during analyte binding [29,30]. Here we demonstrated that similar recognition mechanism can be used for protein detection using SERS-based techniques.’
Q.2. Authors should also provide more meaningful discussions regarding the repeatability and reproducibility of the conducted tests/analysis.
A.2. The following sentences were added:
‘For each experiment, statistical data were obtained on ten SERS substrates. Fifty points were measured for each substrate. The average values and measurement errors for all measurements were calculated.’
‘It can be seen from the data in Figure 6A that SERScont/SERSexp directly depends on the concentration of BoNT A. It should be noted that the concentration of the toxin 1.2 ng/mL is determined in no more than 50% of cases. Based on this, it can be argued that the detection limit of BoNT A for our sensor is 2.4 ng/mL, since the values of SERS intensity for this concentration, taking into account the error, do not overlap with the value of SERS intensity in the control zone.’
Reviewer 2 Report
In this manuscript the authors report on a new sensor of botulinum toxin type A based on a nanostructured silver SERS-substrate functionalized with a modified aptamer. The SERS substrate is characterized by SEM, and the aptamer by UV spectroscopy and circular dichroism. Then the optimal aptamer concentration for sensing is determined as well as the optimal analysis time. Finally the specificityn of the sensor is studied. The limit of detection is 2.4 ng/mL in 1 hour analysis time which is comparable to existing sensors in the literature. The main advantage of the described sensor is it simplicity.
I believe that the manuscript is of potential interest for readers but it is too superficial in the present form. I am surprised that the method is based on Raman detection, but no Raman spectra are shown. It is not clear for me why the spectral window of 1585+-50 cm-1 was used for detection. Finally the lifetime and reusability of the sensors are not discussed.
I also suggest the authos to improve the quality of the figures, in particular the size of the labels should be homogenized between figures.
There are some typos in the text that should be corrected. For instance, line 353, i believe chip is not the correct word and should be replaced by cheap.
Author Response
Q.1. I believe that the manuscript is of potential interest for readers but it is too superficial in the present form. I am surprised that the method is based on Raman detection, but no Raman spectra are shown. It is not clear for me why the spectral window of 1585+-50 cm-1 was used for detection.
A.1. Thank you so much for your comments and recommendations. Figure 4 with Raman spectra was added. Inaccuracies were corrected and changes were made to the text:
- ‘As a result of scanning, maps of the distribution of the intensity of the Raman peak 1,645 cm-1 were recorded.’
- ‘A Raman peak of 1,645 cm-1 (this peak is inherent in the TAMRA label, which is conjugated to the aptamer) was chosen for processing because the maximum plasmon absorption of the SERS substrate used falls on this spectral region [25] and provides maximum amplification of the Raman scattering intensity of this peak.’
Q.2. Finally the lifetime and reusability of the sensors are not discussed.
A.2. The following sentences were added:
‘It should be noted that the shelf life of our sensors is limited only by SERS substrates. In earlier experiments [http://ramanspectr.ru/sers/], it was found that the signal intensity of the test substance on the used sealed substrates is halved after 6 months of storage. Therefore, we believe that its shelf life is six months. The substrates used are disposable, but inexpensive to manufacture (the estimated cost of one sensor is 0.5 USD).’
Q.3. I also suggest the authors to improve the quality of the figures, in particular the size of the labels should be homogenized between figures.
A.3. Figures 3-8 were improved. Inaccuracies in axis labels were fixed.
Reviewer 3 Report
The manuscript by Subekin et al. described an interesting strategy for the SERS detection of botulinum toxin type A. The authors used aptamer for the selective recognition of target analyte. The idea being interesting, the available data provide somewhat weak support to the overall claims. The manuscript is lacking some important details so that repetition of the experiments is difficult. Therefore, I recommend publication of this manuscript after major revisions in Nanomaterials. My special comments are as follows:
1. Abstract is general and needs revision. What are the main achievements in the current work? More quantitative results should be added to Abstract.
2. In the introduction, only antibodies and aptamer SERS-based platforms mentioned. However, their main counterpart molecularly imprinted polymers are missing. Moreover, most of the references are too old. several recent, related works should be referred and cited to further strengthen the research background. For example, about SERS-based sensors: Nature Communications, 2022, 13(1): 5757. And about protein detection by SERS: Biosensors and Bioelectronics, 2021, 174: 112825.
3. Is the lateral resolution important for the detection scheme?
4. How the sensing strategy is sensitive for cross contamination, i.e., when measuring in complex matrices?
5. How the sensor devise is sensitive for mixtures with different concentrated proteins?
6. What is the real-world scenario the authors addressed and which are the relevant matrices? Matrix components might impede the detection, e.g., by blocking binding sites of aptamer.
There are some grammatical errors in the manuscript. Please polish whole manuscript carefully.
Author Response
Q.1. Abstract is general and needs revision. What are the main achievements in the current work? More quantitative results should be added to Abstract.
A.1. Thank you for this comment. We added two sentences about the sensing mechanism and possible area of application as well as several clarifying comments on substrate architecture and type of concentration dependence. The new version of the Abstract:
‘Surface-enhanced Raman scattering (SERS) is a powerful technique for decoding of 2-5-component mixes of analytes. Low concentrations of analytes and complex biological media are cannot been decoded with SERS commonly. Recognition molecules, like antibodies and aptamers, provide an opportunity of specific binding of ultra-low contents of analyte dissolved in complex biological medium. Different approaches have been proposed to provide changes in SERS intensity of an external label upon binding of ultra-low contents of analyte. In this paper, we propose SERS-based aptasensor for the rapid and sensitive detection of botulinum toxin type A. The silver nanoisland SERS substrate was functionalized with Raman-label-modified aptamer. Binding of the target affects the orientation of the label providing changes in analytical signal. This trick allowed detecting botulinum toxin type A in one-stage manner without additional staining with a monotonous dose-dependence and a limit of detection of 2.4 ng/mL in 1 hour. Suggested architecture of the sensor is consistent with the multiarray detection systems for multiplex analysis.’
Q.2. In the introduction, only antibodies and aptamer SERS-based platforms mentioned. However, their main counterpart molecularly imprinted polymers are missing. Moreover, most of the references are too old. several recent, related works should be referred and cited to further strengthen the research background. For example, about SERS-based sensors: Nature Communications, 2022, 13(1): 5757. And about protein detection by SERS: Biosensors and Bioelectronics, 2021, 174: 112825.
A.2. Yes, molecularly imprinted polymers provide an interesting alternative to antibodies and aptamers. We added a reference on protein recognition with molecularly imprinted polymers (the second suggested article) in the Introduction.
Q.3. Is the lateral resolution important for the detection scheme?
A.3. The lateral resolution does not play a role in this case, because we chose an objective with a sufficiently large beam diameter in focus (10 µm), which provided averaging of the signal over a large area. This fact removed the influence of local «hot spots» and we received an average signal.
Q.4. How the sensing strategy is sensitive for cross contamination, i.e., when measuring in complex matrices?
A.4. We agree that cross-specificity is especially significant in practical Implementation of the sensors. Nowadays, we have no biological fluids for multiscale testing of the sensor. So, we used two blood plasma components for preliminary estimation of sensor robustness. Albumin is a main protein component of blood plasma; immunoglobulin G is a component of blood plasma also, but it has the same molecular weight as BoNT. Further experiments are necessary. We added these comments to Discussion part:
‘Here particular components of human blood plasma, namely, serum albumin and immunoglobulin G, were tested to estimate possible effect of non-specific binding that could provide false-positive results. Both proteins did not interfere with detection of BoNT, so the primary experiments revealed no cross-specificity. Further testing of real biological fluids is necessary aiming the practical implementation of the sensors.’
Q.5. How the sensor devise is sensitive for mixtures with different concentrated proteins?
A.5. Using the example of bovine serum, it was shown that a complex matrix has no practical effect on the determination of the toxin, and the sensor developed by us detects BoNT A in simple and complex mixtures with different contents of proteins, low-molecular substances, etc.
A chapter has been added to the text of the article: Determining the influence of the matrix on the definition of BoNT A
Q.6. What is the real-world scenario the authors addressed and which are the relevant matrices? Matrix components might impede the detection, e.g., by blocking binding sites of aptamer.
A.6. During the experiment to determine the toxin in bovine serum, it was found that the components of a complex matrix do not affect the specificity of the definition of BoNT A. Our first data are rather promosing. Further investigation is necessary to study a wide range of different matrices. A subsection has been added to the text of the article: ‘Determining the influence of the matrix on the definition of BoNT A’.
Round 2
Reviewer 2 Report
The authors have taken into consideration my remarks, so I believe the article is now suitable for publication in nanomaterials
overall good, some minor mistakes or typos
Author Response
We are grateful for the evaluation of the work. The language was corrected.
Reviewer 3 Report
The manuscript is suitable for publication now.
Author Response
We are grateful for the evaluation of the work.